# Field Evaluation of Wheat Varieties Using Canopy Temperature Depression in Three Different Climatic Growing Seasons

**DOI:** 10.3390/plants11243471

**Published:** 2022-12-12

**Authors:** Yongmao Chai, Zhangchen Zhao, Shan Lu, Liang Chen, Yingang Hu

**Affiliations:** 1State Key Laboratory of Crop Stress Biology for Arid Areas, College of Agronomy, Northwest A&F University, Xianyang 712100, China; 2Institute of Water Saving Agriculture in Arid Regions of China, Northwest A&F University, Xianyang 712100, China

**Keywords:** canopy structure, photosynthetic parameters, grain yield, drought, spring freezing

## Abstract

During the breeding progress, screening excellent wheat varieties and lines takes lots of labor and time. Moreover, different climatic conditions will bring more complex and unpredictable situations. Therefore, the selection efficiency needs to be improved by applying the proper selection index. This study evaluates the capability of CTD as an index for evaluating wheat germplasm in field conditions and proposes a strategy for the proper and efficient application of CTD as an index in breeding programs. In this study, 186 bread wheat varieties were grown in the field and evaluated for three continuous years with varied climatic conditions: normal, spring freezing, and early drought climatic conditions. The CTD and photosynthetic parameters were investigated at three key growth stages, canopy structural traits at the early grain filling stage, and yield traits at maturity. The variations in CTD among varieties were the highest in normal conditions and lowest in spring freezing conditions. CTD at the three growing stages was significantly and positively correlated for each growing season, and CTD at the middle grain filling stage was most significantly correlated across the three growing seasons, suggesting that CTD at the middle grain filling stage might be more important for evaluation. CTD was greatly affected by photosynthetic and canopy structural traits, which varied in different climatic conditions. Plant height, peduncle length, and the distance of the flag leaf to the spike were negatively correlated with CTD at the middle grain filling stage in both normal and drought conditions but positively correlated with CTD at the three stages in spring freezing conditions. Flag leaf length was positively correlated with CTD at the three stages in normal conditions but negatively correlated with CTD at the heading and middle grain filling stages in spring freezing conditions. Further analysis showed that CTD could be an index for evaluating the photosynthetic and yield traits of wheat germplasm in different environments, with varied characteristics in different climatic conditions. In normal conditions, the varieties with higher CTDs at the early filling stage had higher photosynthetic capacities and higher yields; in drought conditions, the varieties with high CTDs had better photosynthetic capacities, but those with moderate CTD had higher yield, while in spring freezing conditions, there were no differences in yield and biomass among the CTD groups. In sum, CTD could be used as an index to screen wheat varieties in specific climatic conditions, especially in normal and drought conditions, for photosynthetic parameters and some yield traits.

## 1. Introduction

In recent years, climate change has resulted in significant variations in seasonal climatic conditions within growing seasons with the emergence of abnormal weather events, including drought, heat, spring freezing, and others, which have had a great impact on the growth and production of many crops [1,2,3,4,5]. It was shown that within the range of 15.8–27.7 °C, for every 1 °C increase in the daily average temperature during the wheat grain filling period, the filling period will be shortened by 3.1 days, and the 1000-kernel weight will be reduced by 2.8 mg. A 1 °C increase in the global average temperature would reduce wheat yields by 5.7% in the future based on model estimation. Therefore, accelerating the breeding of wheat varieties adapted to the future climate is essential to ensure world food security [6]. In the past two decades, the wide application of genome sequencing and molecular markers in plant breeding has made it possible to study plant genetic variation at the genome level. The combination of genotype and phenotype data has great potential to accelerate plant breeding development. However, the development of phenotype techniques still faces enormous challenges, particularly for complex traits such as drought, heat, and salt resistance [7,8]. Therefore, developing direct or indirect screening methods that can detect genetic variations in populations with low genotype–environment interactions and a limited environmental influence is critical [8].

Canopy temperature (CT) has been widely employed to evaluate and select wheat in drought and hot conditions as a valuable predictor of abiotic stresses [8,9,10]. CT integrates the main canopy morphological characteristics, such as leaf color, leaf size, spike size, inflorescence length, plant height, etc., so CT is greatly affected by the canopy structure [11]. CT measured before and after flowering is significantly correlated with plant height, as the dwarfing alleles of *Rht-B1b* and *Rht-D1b* are positive for CT, according to QTL analyses in multiple environments [12]. CT is also used as an indicator of stomatal characters and photosynthetic capacity in suitable conditions [13]. CT has been proposed as a significant indicator for measuring agricultural responses to water shortages, high temperatures, and other environmental stresses, and it has been used to evaluate the drought resistance of several crops, including sorghum, soybean, wheat, and rice [14,15]. CT is negatively correlated with yield, mostly in drought and thermal conditions, and a cooler canopy is beneficial to yielding gains in these stresses [16,17].

Derived from CT, canopy temperature depression (CTD) has been suggested to reflect the difference between the plant canopy temperature (Tc) and the ambient air temperature (Ta) [18]. CTD has been used as a selection criterion for improving drought and heat resistance in increasing the yield of CIMMYT wheat varieties [19,20]. CTD was positively correlated with grain yield and leaf area index in heat stress [21]. The correlation between CTD and stomatal conductance is variable in different climatic conditions [13]. Along with the development of high-throughput field phenotyping, CT can be measured with an infrared camera carried by unmanned aerial vehicles (UAV) in field experiments [8], which is a simple and reliable method for estimating abiotic stresses in plants [22,23] and can greatly increase the repeatability of CT [24].

However, different climatic conditions will bring complex and unpredictable situations to wheat breeding, and the reliability of CTD as a selection index has been questioned [25]. Changes in climatic conditions significantly impact wheat growth, which creates great challenges for the evaluation of germplasm in field conditions. The climatic conditions in a single region may vary greatly between growing seasons. As observed in this study, three different climatic conditions (normal, spring freezing, and early drought) occurred in three continuous growing seasons. The hypothesis of this study is that in different climatic conditions CTD at the key growth stages could be used as an index for evaluating the photosynthetic and yield characteristics of a large amount of wheat germplasm, but what it reflects might vary in different conditions. Therefore, to clarify the capability of CTD as an index in evaluating the canopy structural traits, photosynthetic traits, and yield traits of wheat under changing climatic conditions and to improve the selection efficiency in the wheat breeding program, 186 bread wheat varieties were grown in the same field for three consecutive growing seasons. CTD and photosynthetic parameters at three key stages of heading—the early and middle grain filling stage, the canopy structure at the early grain filling stage, and the yield traits at the mature stage—were investigated. The relationship between CTDs at three key stages and with canopy structural traits, photosynthetic parameters, and yield traits was studied. The influencing factors of CTD and its utilization as a selection index were analyzed. It may make up for the gaps in the application of CTD as an index in various environments.

## 2. Plant Materials and Methods

### 2.1. Plant Materials and Field Planting

The test materials included 186 bread wheat varieties or advanced lines, mainly from winter wheat regional trials in the Huang–Huai Winter Wheat Region of China. Among them, 75 varieties passed the national trials, 42 varieties passed the provincial trial for release in the corresponding regions, and 7 wheat varieties introduced from overseas were included, as shown in Appendix A.

### 2.2. Planting and Management of Field Trials

The field experiment was conducted in the experimental field of the Institute of Water Saving Agriculture in Arid Regions of China, Northwest A&F University, Yangling, Shaanxi (34°7′ N, 108°3′42″ E) during three growing seasons (October to early June of 2016–2017, 2017–2018, and 2018–2019). Wheat was sown in early October and harvested in early June of each growing season. The experiment was conducted in an incomplete randomized block design with 2 replicates. The plots were 3.0 m^2^ in area, with 6 rows of 2.0 m in length, row spacing of 25 cm, and plant spacing of 3.3 cm. Based on local practices in wheat production, 750 kg/Ha compound fertilizers (N:P:K = 18:18:5) were applied before sowing during the preparation of the land, which was the same for each year. Wheat growth mainly depended on soil water storage and precipitation during the growth period. Chemical herbicides were used for field weeding in mid-November. Fungicides were used in late April to control stripe rust and powdery mildew, and other field management procedures were the same as local production.

### 2.3. Determination of Canopy Temperature

At the stages of heading (GS55), early filling (GS71), and middle filling (GS75) of wheat, CTs (Tc) were measured using a handheld portable infrared thermal imager (Variocam Hr, Infratec, Dresden, Germany). Meanwhile, a Huayi PM6508 calorimeter was used to measure the air temperature (Ta) every 20 min. The growth stages were recorded for each variety, and the difference in the heading time of these varieties was about 3–5 days. When about 50% of tillers in a variety plot reached the growth stage, CT measurements were carried out thrice with a handheld portable infrared thermal imager from 12:00 to 15:00 on a sunny day. The near-infrared images were taken twice, one plot at a time, on the middle 3 rows of a 1-m-long plot by standing on the same side of the plot, tilted horizontally at an angle of approximately 30°. Five areas, including wheat leaves, stems, and spikes, were chosen from the images, and the canopy temperature was captured using the IRBIS^®^ View-Infrarot Anzeiges software along with other equipment. Then, the CTD (°C) was calculated following the equation CTD = Ta − Tc.

### 2.4. Determination of Canopy Structural Traits

At the middle grain filling stage (GS75), a ruler was used to measure the spike layer width, spike length, distance from the spike to the flag leaf, peduncle length, plant height, and the length and width of the flag leaf using the standard methods [26].

### 2.5. Determination of Photosynthetic Parameters

At the heading, early filling, and middle filling stages of the wheat, at the time of taking near-infrared images, the photosynthetic parameters of flag leaves were investigated, which took about 3 days for these varieties according to their heading date. The photosynthetic parameters of flag leaves were measured with a Li-6400XT gas exchange system (Li-Cor Inc., Lincoln, NE, USA) from 8:30 to 13:00 on sunny days, including the net photosynthetic rate (A), stomatal conductance (Gs), transpiration rate €, and intercellular CO_2_ concentration (Ci). A 2 × 3 cm^2^ leaf chamber was used in the fixed conditions of photosynthetic active radiation (1200 mmol m^−2^ s^−1^), airflow rate (0.5 dm^3^ min^−1^), leaf chamber temperature (25 °C), and CO_2_ (390–410 mmol mol^−1^) so as to avoid the negative effects of weather conditions during the day. Instantaneous water-use efficiency (WUEi) was calculated as the ratio of assimilation to transpiration (A/E). The Ci/Ca ratio was calculated to reflect the ability of stomata to limit CO_2_.

### 2.6. Determination of Yield Traits

At the maturity stage (GS91), 1 m^2^ in the center rows of each plot was harvested from the ground, dried in the greenhouse for 1 week, weighed to obtain biomass (including leaves, stems, and chaff), then threshed and dried to about 13% water content so as to obtain grain yield, and the harvest index was calculated [26].

### 2.7. Data Analysis

Data analysis—including descriptive statistics, correlation analysis with Pearson’s correlation, cluster analysis with the Ward method, and one-way ANOVA analysis to test for the significance of differences in the traits among different groups using Duncan’s new multiple-range tests—was conducted using Statistics for Windows, Version 17.0. (SPSS Inc., Chicago, IL, USA).

## 3. Results

### 3.1. Climatic Conditions of the Three Growing Seasons

There were some variations in the air temperatures over the three wheat growing seasons, which were 9.01 °C, 9.53 °C, and 8.44 °C. It was important to note that the lowest temperature suddenly dropped to −0.5 °C on the night of 7 April 2018, which caused severe freezing damage to most wheat varieties at the booting–heading stage. The main stem ears and several tillers of many varieties suffered severe freezing damage; therefore, 2017–2018 was considered a spring freezing climatic condition. The rainfall amounts for the three growing seasons were 288.1 mm, 324.8 mm, and 189.3 mm, which, for the 2016–2017 and 2017–2018 growing seasons, were greater than the average of the last 10 years (219.7 mm) for the region, while for 2018–2019, it was less, with only 55.9 mm rainfall in the early growing season (October to March) compared with 184.2 mm and 201.5 mm for the 2016–2017 and 2017–2018 growing season, respectively, resulting in severe drought at the earlier growth stage. Thus, the 2018–2019 growing season was characterized as an early drought prior to reproductive growth, as shown in Figure 1. Therefore, the climatic conditions of the three growing seasons could be considered normal (2016–2017), spring freezing (2017–2018), and early drought (2018–2019).

### 3.2. Variations in CTD among the Wheat Varieties

There were wide variations in CTD among the wheat varieties across the three growing seasons. In the normal (2016–2017) growing season, CTD variations were more obvious among genotypes, with similar trends observed among varieties at the three key growth stages, while in the spring freezing (2017–2018) growing season, the overall variations in CTD at the heading and early filling stages were lower; in the drought (2018–2019) growing season, the greatest number of CTD variations were found at the heading stage, and the smallest were at the early filling stage, as shown in Figure 2 and Appendix A.

CTDs at the three stages were significantly and positively correlated in the same growing season, with the highest correlations (0.75) observed in the normal (2016–2017) growing season, followed by the drought (2018–2019) growing season and the lowest (0.382) in freezing (2017–2018) growing season, as shown in Table 1. The correlations between CTDs at the heading stage and the early grain filling stage were higher than between CTDs at the early grain filling stage and the middle grain filling stage in all three growing seasons. Across the three growing seasons, the CTD at the middle grain filling stage of the drought growing season was significantly and positively correlated with the CTDs at the three stages of the normal growing season (0.199, 0.150, 0.259), but it was significantly and negatively correlated with the CTDs at the three stages of the freezing growing season (−0.273, −0.173, −0.388).

### 3.3. CTD and Canopy Structure

The canopy structure traits, including the width of the spike layer, spike length, the length and width of flag leaves, flag leaf area, and plant height, were investigated. In Figure 3A, freezing greatly reduced plant height (22.30%), flag leaf length (27.9%), and flag leaf area (13.5%) and shortened peduncle length (31.7%), while drought reduced the distance from spike to leaf (30.8%) the most, as well as flag leaf length (16.9%) and plant height (13.4%).

A correlation analysis between the CTDs at the three growth stages and the canopy structure traits revealed various correlations across the three growing seasons, as shown in Figure 4. In the normal growing season, the CTDs at all three growth stages were significantly and positively correlated with flag leaf length (0.231, 0.227, 0.238); the CTD at the heading stage was negatively correlated with the peduncle length (−0.156) and the distance from the spike to the leaf (−0.189); and the CTD at the middle filling stage was negative correlated with plant height (−0.207), the peduncle length (−0.225), and distance from the spike to the leaf (−0.228).

In the spring freezing growing season, the CTDs at the three growth stages were all positively correlated with plant height (0.462, 0.23, 0.379), the peduncle length (0.336, 0.209, 0.237), and the distance from the spike to the leaf (0.232, 0.168, 0.156), and the CTDs at both the heading and middle filling stages were negatively correlated with the flag leaf length (−0.227, −0.156).

In the drought growing season, the CTDs at the three growth stages were all positively and significantly correlated with the flag leaf length (0.422, 0.198, 0.366) and flag leaf area (0.457, 0.209, 0.275); the CTD at both the heading and early filling stages was positively and significantly correlated with flag leaf width (0.342, 0.152); and the CTD at the early filling stage was positively correlated with plant height (0.289) and the distance from the spikes to the leaves (0.159); however, the CTD at the middle filling stage was negatively correlated with the plant height (−0.341) and the distance from the spikes to the leaves (0.173).

Clearly, there were more canopy structural traits correlated with the CTDs at the three stages in the drought and freezing growing seasons, which suggested that canopy structure traits were the main traits affecting CTD, especially in the stressed growing seasons.

### 3.4. CTD and Photosynthetic Traits

The investigation of the photosynthetic parameters indicated there was a wide range of variations among the wheat varieties in the three growing seasons, as shown in Table 2. Higher A, E, and Gs for flag leaves were observed in the drought growing season, while they were lowest in the freezing growing season. The A/E ratios of flag leaves in freezing and drought growing seasons were all lower than in the normal growing season.

The correlations between CTDs and photosynthetic parameters at the corresponding stages were further investigated, and various correlations were observed in different growing seasons, as shown in Figure 5. In the normal growing season, the CTD was positively correlated with the A/E (0.240) and Ci/Ca (0.156) but negatively correlated with E (−0.264) at the heading stage; the CTD was positively correlated with A (0.289) and E (0.337) at the early grain filling stage; and the CTD was positively correlated with E (0.184), Ci (0.501), Gs (0.248), and Ci/Ca (0.518) but negatively correlated with A (−0.416) and A/E (−0.243) at the middle grain filling stage.

In the freezing growing season, the CTD was negatively correlated with A (−0.152), E (−0.281), and Gs (−0.251) but positively correlated with A/E (0.189) at the heading stage; the CTD was negatively correlated with E (−0.281) but positively correlated with A/E (0.177) at the early grain filling stage; and the CTD was negatively correlated with Ci (−0.149) and Gs (−0.185) at the middle grain filling stage.

In the drought growing season, the CTD was positively correlated with A (0.585), E (0.533), Ci (0.558), Gs (0.631), and Ci/Ca (0.559) at the heading stage; the CTD was negatively correlated with A (−0.306) but positively correlated with Ci (0.195) and Ci/Ca (0.197) at the early filling stage; and the CTD was positively correlated with A (0.234), E (0.231), and Gs (0.192) but negatively correlated with A/E (−0.164) at the middle filling stage. This suggested that CTD was greatly affected by the photosynthetic performance of the plant and could be used to reflect the status of photosynthetic capability, especially in normal and drought growing seasons.

### 3.5. CTD and Yield Traits

Compared with the normal growing season, the grains per spike and spike number per m^2^ decreased by 17.80% and 3.60% and 49.89% and 38.36%, while the 1000-kernel weight increased by 24.11% and 23.56% in the freezing and drought growing seasons, respectively. Finally, the grain yield and biomass decreased by 20.62% and 15.51% and 30.84% and 31.45% in the freezing and drought growing seasons, respectively, as shown in Figure 3B.

As shown in Table 3, a correlation analysis indicated that, in the normal growing season, the CTD at the heading stage was positively correlated with 1000-kernel weight (0.234), the CTD at the early filling stage was negatively correlated with the spike number per m2 (−0.193) and biomass (−0.238), and the CTD at the three growth stages was all positively correlated with the harvest index (0.232, 0.184, 0.147).

In the spring freezing growing season, CTDs at both the heading and early filling stages were only negatively correlated with the harvest index (−0.218, −0.155).

In the drought growing season, CTDs at both the heading and middle filling stages were negatively correlated with grains per spike (−0.279, −0.284) but positively correlated with the harvest index (0.198, 0.278); the CTD at the middle filling stage was negatively correlated with spike number per m^2^ (−0.165), while the CTD at the early filling stage was positively correlated with biomass (0.150).

### 3.6. Comparisons among Clusters of Varieties

To further understand the effects of CTD and canopy structural traits on yield traits, three kinds of cluster analysis were conducted using CTDs at the three growth stages, the canopy structural traits, and their combination for each growing season using the Ward method due to the greater influences of canopy structural traits on CTD. For easy comparison, the wheat varieties were clustered into three groups in each climatic condition. The clusters are presented in Appendix A. The results of clustering based on the CTDs at the three growth stages and the canopy structural traits all showed significant differences in CTDs at the three stages (Appendix A), canopy structural traits (Appendix A), most of the photosynthetic parameters (Figure 6, Appendix A), and some yield traits among the clusters of varieties (Appendix A) in the three different growing seasons. This only showed the results of clustering based on the combination of the CTDs at the three growth stages and the canopy structural traits. Significant differences in the CTD values and canopy structural traits were observed among the three clusters of wheat varieties, as shown in Table 4. In general, the varieties could be classified as higher or lower CTDs at all three stages, with varied CTDs at different growth stages for different growing seasons. Furthermore, significant differences in the canopy structural traits were also observed, especially in plant height and flag leaf-related traits.

Multiple comparisons of the yield traits were further conducted, as shown in Table 5 and Figure 6. In the normal growing season, wheat varieties with lower CTDs at the three stages were placed in Cluster 1, and those with higher CTDs at the three stages were placed in Cluster 2 and Cluster 3. Significant differences in grain yield and spike number per m^2^, 1000-kernel weight, and the harvest index were observed between Cluster 1 and Cluster 3, but not between Cluster 2 and Cluster 3.

In the spring freezing growing season, there were significant differences in CTDs at the heading and middle grain filling stages among the three clusters. However, differences were only found in the spike number per m^2^ and the 1000-kernel weight among the three clusters.

In the drought growing season, varieties in Cluster 2 with moderate CTDs at heading and early grain filling stages and higher plant height obtained the highest grain yield, while significant differences were observed in the biomass and spike number per m^2^ between varieties in Cluster 2 and the other two clusters.

In normal conditions, wheat varieties with higher CTDs achieve higher grain yields and 1000-kernel weights; in drought conditions, the wheat varieties with moderate CTDs achieve higher grain yields, biomasses, and spike numbers per m^2^; in spring freezing, there were no differences in yield and biomass among the CTD groups.

## 4. Discussions

### 4.1. CTD Variation at Different Growth Stages and Climatic Conditions

CTD is highly variable at different growth stages and in various growing conditions. The coefficient of variation in CTDs averages 39% in pre-heading drought conditions and 25% in the normal growing season [27]. The CTD range (−4~8 °C) is greater in the drought growing season than in the rainy growing season (−1~4 °C) [28]. In this study, the CTD range was larger during the normal growing season, followed by the drought growing season, while it was the smallest in the spring freezing growing season, with great variations at the heading and middle grain filling stages. After spring freezing, the newly developed tillers were smaller, and the canopy closed poorly, resulting in lower CTDs, which reduces the variations in CTDs among wheat varieties. CTD can be correlated in different growth periods and can maintain a high correlation with green status even if there is no transpiration tissue [23]. In this study, there were various correlations between CTDs measured at the three growth stages in the three different climatic growing seasons, with the strongest in the normal growing season (R = 0.750), followed by in the drought growing season (R = 0.616), and the weakest was in the freezing growing season (R = 0.382). The plants were severely injured in stress conditions, and the canopy could not be closed well, which also increased the apparent error in the CTD measurement. The range of CTDs and the correlation of different growth stages reflect the stability of the canopy, which can be used as a reference to construct a reasonable canopy.

### 4.2. CTD and Canopy Structure

The CTD reflects the canopy function and was greatly affected by the traits contributing to the canopy structure. Previous studies observed that CT was strong and negatively correlated with plant height in common wheat in drought conditions (R = −0.64) [15,27], and CT was negatively correlated with peduncle length [21]. In this study, various correlations were observed between CTDs and canopy structural traits in different conditions; CTD was positively correlated with flag leaf length and negatively correlated with peduncle length and the distance from the spikes to the leaves in normal conditions; CTD was positively correlated with all flag leaf traits in drought conditions; and CTD was positively correlated with plant height, peduncle length, and distance from the spikes to the leaves and negatively correlated with flag leaf length in spring freezing conditions. Spring freezing severely damaged spikes and flag leaves, further reducing plant height and related traits [29], as observed in Figure 3, which was positively correlated with the CTDs at the three stages. Flag leaf-related traits were seriously affected by early drought, which were all positively correlated with the CTDs at the three stages. Therefore, the canopy structural traits were the most important factors affecting CTD, and CTD could reflect the main canopy traits in various climate conditions.

### 4.3. CTD and Photosynthetic Traits

Previous studies found that CTD could indirectly reflect the photosynthetic properties of plants, especially E at the whole crop level, and CTD was related to Gs and A in wheat [30]. In the present study, various correlations were observed for the photosynthetic parameters with CTD in different climate conditions. In normal conditions, CTD was either positively or negatively correlated with A and E at the early and middle grain filling stages. In early drought conditions, CTD was strongly and positively correlated with most of the photosynthetic parameters at the heading stage; this was consistent with the expectation that Gs is the main factor determining the genotype differences of CTD in drought-prone environments [31]. Flag leaves and spikes were severely damaged, and photosynthetic capacity was reduced after spring freezing damage [32]. At the heading and early filling stages, A was lower than in normal and drought conditions, while Ci and Ci/Ca were on the contrary, as shown in Table 2, indicating that freezing injuries damaged cell membranes and affected CO_2_ transport. CTD and some photosynthesis parameters were strongly correlated at the heading stage, but less at the early and middle grain filling stages, as shown in Figure 5. This suggests that CTD can be used as an index to evaluate photosynthetic traits in different climatic conditions, as in Figure 7.

### 4.4. CTD and Yield Traits

CT’s ability to screen wheat genotypes under water stress is based on its close relationship with grain yield and genotype differences among varieties [31,33]. Previous studies have shown various correlations between CT at different stages and some yield traits and found a substantial link between cooler CTs during the grain filling stage and yield, with both positive and negative correlations observed in different situations [11,12,17,34,35]. Low canopy temperatures before anthesis are associated with an increase in biomass, grain number, and yield, but low canopy temperatures after anthesis were not related [12]. Multiple QTL analyses also observed that the QTLs for CT were pleiotropic with yield, biomass, and other characteristics [15,34,35]. In this study, CTD was less correlated with yield traits in the three climatic conditions; this might be due to the involvement of many diverse wheat varieties. Similar results showed that the relationship between CTDs and yield traits is weak in favorable conditions [15]. Wheat varieties were further clustered together into groups based on CTDs and canopy structural traits at the three growth stages, as plant height and flag leaf-related traits had a strong influence on CTD. As shown in Table 5, Figure 7, and Appendix A, in normal conditions, wheat varieties with higher CTDs (cool canopy) at the three stages achieved the highest grain yield; the varieties with the lowest CTDs at the heading and middle grain filling stages and taller plants achieved the lowest grain yields and harvest indexes. In drought conditions, the wheat varieties with moderate CTDs achieved higher grain yields, biomasses, and spike numbers per m^2^. In spring freezing, there was no difference in yield and biomass. These results were partially consistent with the yield benefit of a cooler canopy at the grain filling stage in drought and high temperature conditions [16,17], and higher CTDs (cooler canopy) can be used as a selection factor to promote drought and heat tolerance in bread wheat [21,33,36,37,38]. These results suggest that CTD can be applied as a selection index for wheat yields in different climatic conditions. The canopy structural traits would be combined with CTD to select yield traits. As mentioned earlier, the polygenic control, the environmental sensitivity of CT, and the value of CTD for wheat variety selection should be thoroughly elucidated [12].

In general, CTD could be an index for evaluating wheat germplasm for canopy structural traits, photosynthetic capability, and yield traits in the field under different climatic conditions, especially for normal and drought conditions, and the application of CTD in freezing conditions should be further investigated.

## 5. Conclusions

Based on this study, it was found that CTD could be an index for evaluating the photosynthetic and yield traits of wheat germplasm in different environments, with varied characteristics in different climatic conditions. In normal conditions, the varieties with higher CTDs at the three growth stages had higher photosynthetic capacities and the highest yields; in drought conditions, the varieties with high CTDs had better photosynthetic capacity, but those with moderate CTD had the highest yield; and in spring freezing conditions, though CTD was hard on yield traits, it still could reflect canopy structural traits and the photosynthetic capacity to some extent.

## Figures and Tables

**Figure 1 plants-11-03471-f001:**
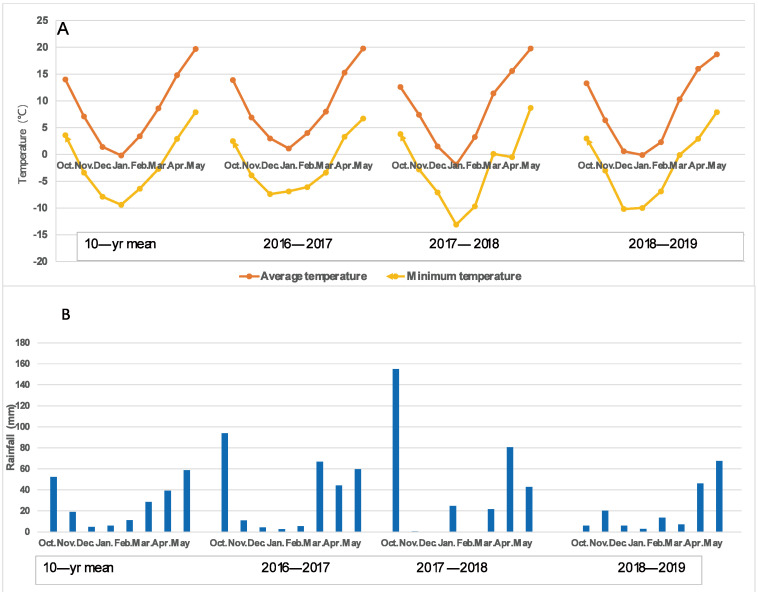
The average and minimum air temperature (**A**) and rainfall (**B**) during the three wheat growing seasons. A lower temperature of −0.5 °C occurred on the night of 7 April 2018, which caused serious spring freezing damage to the wheat varieties at the booting stage. The rainfall amount in the early stage (October to March) of the 2018–2019 growing season was only 55.9 mm, while it was 184.2 mm and 201.5 mm for the 2016–2017 and 2017–2018 growing seasons, respectively.

**Figure 2 plants-11-03471-f002:**
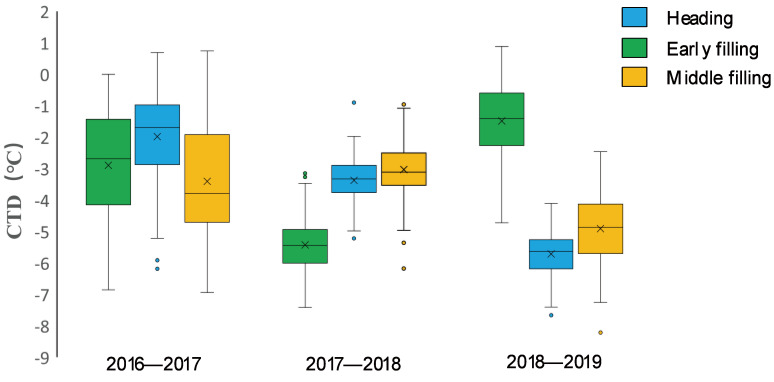
The variations in the CTD value of wheat varieties at the three growth stages of the three wheat growth seasons are shown in boxplot diagrams. CTD variations were more obvious among wheat varieties, with similar trends observed among wheat varieties at the three key growth stages in the 2016–2017 (normal) growing season, while the overall variations in CTD in the heading and early filling stages were smaller in the 2017–2018 (spring freezing) growing season. The greatest number of variations in CTD were at the heading stage, and the smallest were at the early filling stage in the 2018–2019 (early drought) growing season.

**Figure 3 plants-11-03471-f003:**
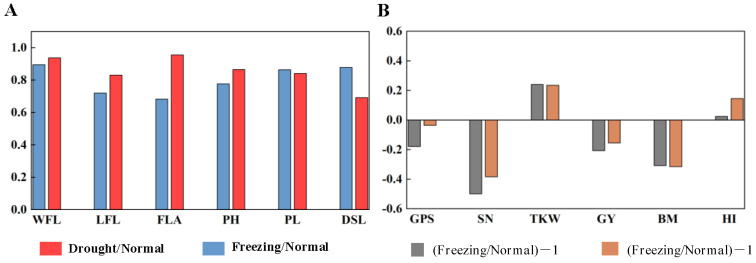
The effects of spring freezing and early drought on canopy structural traits (**A**) and yield traits (**B**) compared with the normal growing season. WFL: width of flag leaf; LFL: length of flag leaf; FLA: flag leaf area; PH: plant height; PL: peduncle length; DSL: distance from spike to leaves; GPS: grains per spike; SN: spike number per m^2^; TKW: 1000-kernel weight; GY: grain yield; BM: biomass aboveground; HI: harvest index. Freezing/Normal, Drought/Normal: the ratio of the mean value for canopy structural traits under freezing and drought conditions to that under normal conditions, respectively; (Freezing/Normal)-1, (Drought/Normal)-1: the ratio of the mean value for yield traits under freezing and drought conditions to that under normal conditions minus 1, respectively.

**Figure 4 plants-11-03471-f004:**
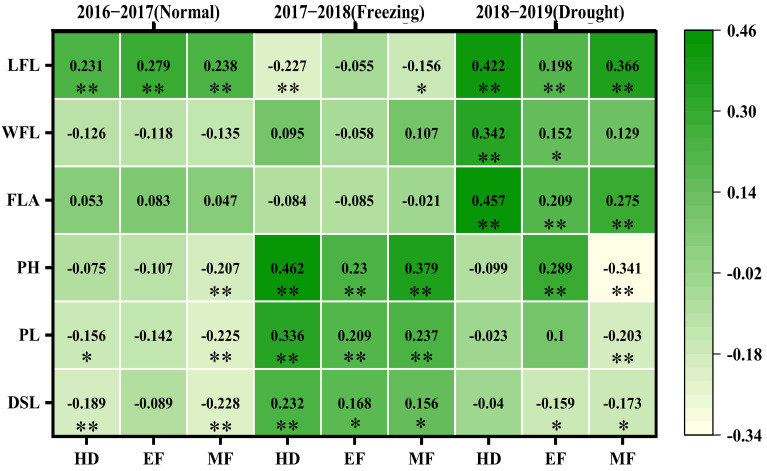
The correlation coefficients between CTDs at three growth stages with the canopy structural traits of three growing seasons. The correlations were estimated using the Pearson correlations from the SPSS; * and ** indicate the correlations significant for *p* < 0.05 and *p* < 0.01, respectively. WFL: width of flag leaf; LFL: length of flag leaf; FLA: flag leaf area; PH: plant height; PL: peduncle length; DSL: distance from spike to leaves; HD: heading; EF: early filling; MF: middle filling.

**Figure 5 plants-11-03471-f005:**
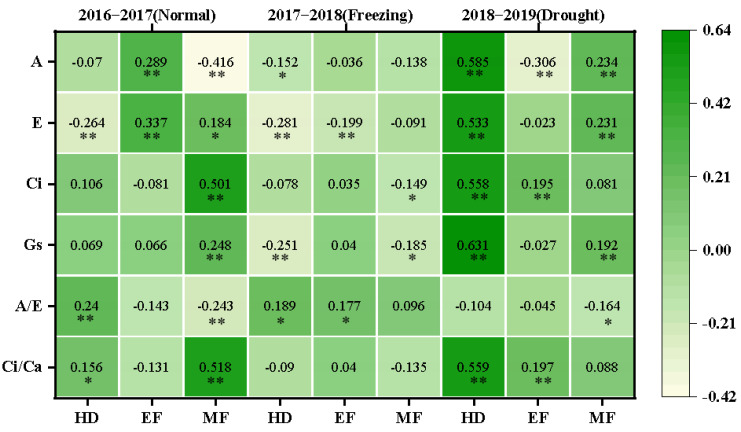
The correlation coefficients between CTDs at three growth stages with photosynthetic parameters at the corresponding stages in the three growing seasons. The correlations were estimated using the Pearson correlations from the SPSS; * and ** indicate the correlations significant for *p* < 0.05 and *p* < 0.01, respectively. A: net photosynthetic rate; E: transpiration rate; Gs: stomatal conductance; Ci: intercellular CO_2_ concentration; A/E: instant water use efficiency; Ci/Ca: stomatal limitation ratio; HD: heading; EF: early filling; MF: middle filling.

**Figure 6 plants-11-03471-f006:**
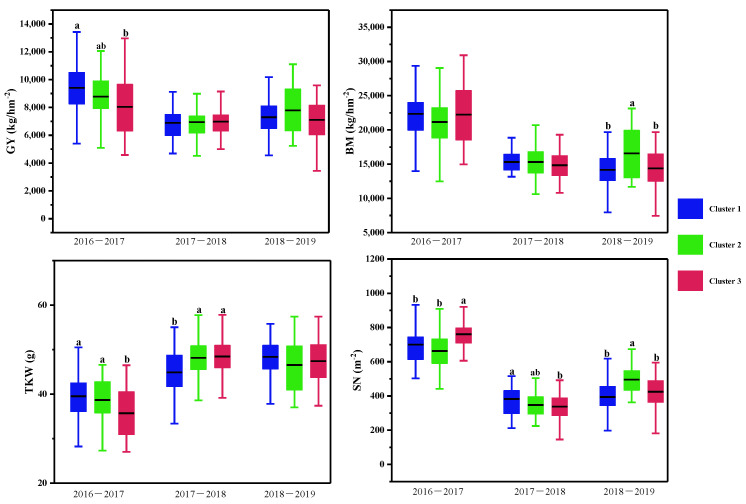
Multiple comparisons of yield traits among the three clusters based on the CTD at three growth stages and canopy structural traits for each growing season. The clustering was conducted with the CTD at the three growth stages and structural traits for each growing season using the Ward method. Then multiple comparisons were conducted among different clusters using the Duncan method, different lowercase letters indicated the differences significant at the *p* < 0.05 level. SN: spike number per m^2^; TKW: 1000-kernel weight; GY: grain yield; BM: biomass.

**Figure 7 plants-11-03471-f007:**
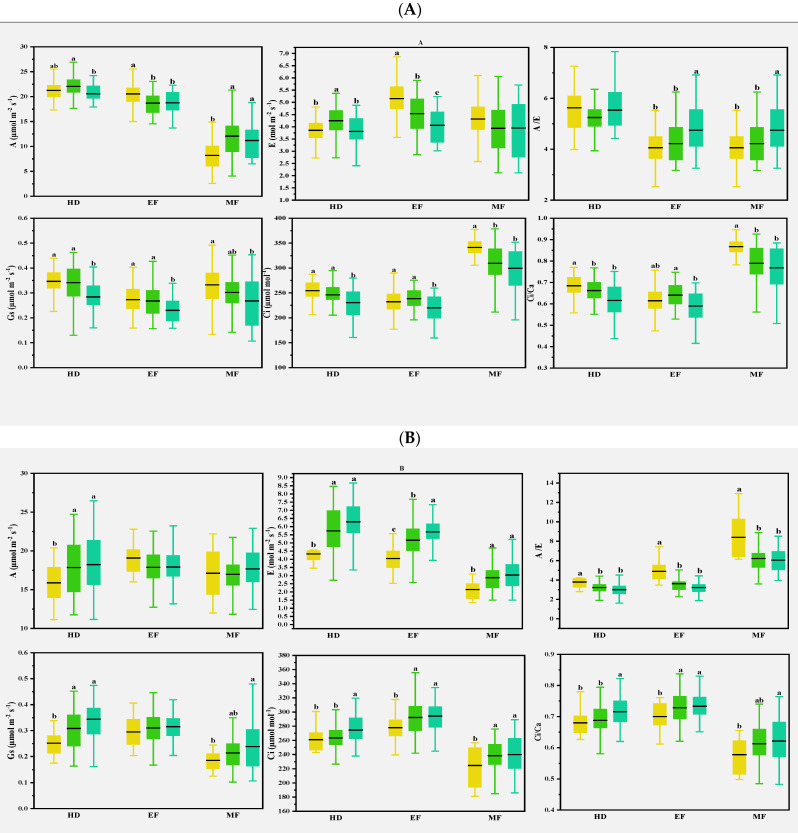
Multiple comparisons of photosynthetic traits among the three clusters based on the CTDs at the growth stages and canopy structural traits of the 2016–2017 (**A**), 2017–2018 (**B**), and 2018–2019 (**C**) growing seasons. The clustering was conducted with the CTDs at the three growth stages and structural traits for each growing season using the Ward method. Then, multiple comparisons were conducted among different clusters using the Duncan method. Different lowercase letters indicate differences significant at the *p* < 0.05 level. A: net photosynthetic rate; E: transpiration rate; Gs: stomatal conductance; Ci: intercellular CO_2_ concentration; A/E: instant water use efficiency; Ci/Ca: stomatal limitation ratio. HD: heading; EF: early filling; MF: middle filling.

**Table 1 plants-11-03471-t001:** Correlation coefficients of CTDs between the three growth stages of the three growing seasons.

Growing Seasons		2016–2017 (Normal)			2017–2018 (Freezing)			2018–2019 (Drought)		
		HD	EF	MF	HD	EF	MF	HD	EF	MF
2016–2017 (Normal)	HD	1	0.704 **	0.750 **	−0.093	0.04	−0.106	0.062	0.012	0.199 **
	EF		1	0.647 **	−0.132	0.07	−0.068	0.079	0.028	0.150 *
	MF			1	−0.14	−0.023	−0.189 **	0.124	0	0.259 **
2017–2018 (Freezing)	HD				1	0.372 **	0.382 **	−0.123	0.210 **	−0.278 **
	EF					1	0.313 **	−0.028	−0.105	−0.173 *
	MF						1	−0.197 **	−0.059	−0.388 **
2018–2019 (Drought)	HD							1	0.547 **	0.616 **
	EF								1	0.284 **
	MF									1

Note: The correlations were estimated using the Person correlations from the SPSS; * and ** indicate the correlations significant for *p* < 0.05 and *p* < 0.01 level, respectively. HD: heading; EF: early grain filling; MF: middle grain filling.

**Table 2 plants-11-03471-t002:** Photosynthetic parameters at three growth stages of the three growing seasons.

Growing Season		A (µmol m^−2^ s^−1^)		E (mol m^−2^ s^−1^)		Ci (µmol mol^−1^)	
	Stages	Mean ± SD	CV	Mean ± SD	CV	Mean ± SD	CV
2016−2017 (Normal)	HD	21.47 ± 2.29	10.67%	4.01 ± 0.62	15.46%	247.88 ± 23.39	9.44%
	EF	19.55 ± 2.39	12.23%	4.75 ± 0.84	17.68%	233.1 ± 23.69	10.16%
	MF	10.17 ± 4.38	43.07%	4.11 ± 0.93	22.63%	322.71 ± 36.36	11.27%
2017−2018 (Freezing)	HD	17.84 ± 3.51	19.67%	5.85 ± 1.46	24.96%	267.98 ± 19.32	7.21%
	EF	18.02 ± 2.23	12.38%	5.28 ± 1.07	20.27%	291.75 ± 23.24	7.97%
	MF	17.31 ± 2.43	14.04%	2.88 ± 0.8	27.78%	237.87 ± 23.91	10.05%
2018−2019 (Drought)	HD	27.5 ± 4.05	14.73%	8.32 ± 1.58	18.99%	259.25 ± 49.19	18.97%
	EF	27.67 ± 4.22	15.25%	7.37 ± 1.53	20.76%	284.43 ± 17	5.98%
	MF	22.38 ± 2.19	9.79%	4.72 ± 0.91	19.28%	270.6 ± 19.31	7.14%
**Growing Season**		**Gs (µmol m^−2^ s^−1^)**		**A/E**		**Ci/Ca**	
	**Stages**	**Mean ± SD**	**CV**	**Mean ± SD**	**CV**	**Mean ± SD**	**CV**
2016−2017 (Normal)	HD	0.34 ± 0.07	20.59%	5.46 ± 0.81	14.84%	0.66 ± 0.06	9.09%
	EF	0.26 ± 0.06	23.08%	4.23 ± 0.79	18.68%	0.62 ± 0.06	9.68%
	MF	0.31 ± 0.08	25.81%	4.23 ± 0.79	18.68%	0.82 ± 0.09	10.98%
2017−2018 (Freezing)	HD	0.32 ± 0.07	21.88%	3.17 ± 0.67	21.14%	0.7 ± 0.05	7.14%
	EF	0.31 ± 0.05	16.13%	3.43 ± 0.82	23.91%	0.73 ± 0.05	6.85%
	MF	0.22 ± 0.07	31.82%	6.11 ± 1.13	18.49%	0.62 ± 0.07	11.29%
2018−2019 (Drought)	HD	0.42 ± 0.16	38.10%	3.34 ± 0.42	12.57%	0.66 ± 0.12	18.18%
	EF	0.47 ± 0.1	21.28%	3.83 ± 0.51	13.32%	0.71 ± 0.04	5.63%
	MF	0.34 ± 0.07	20.59%	4.91 ± 0.84	17.11%	0.69 ± 0.05	7.25%

Note: The data are shown as the mean ± standard deviation; CV: coefficients of variations (CV); A: net photosynthetic rate; E: transpiration rate; Gs: stomatal conductance; Ci: intercellular CO_2_ concentration; A/E: instant water use efficiency; Ci/Ca: stomatal limitation ratio. HD: heading; EF: early filling; MF: middle filling.

**Table 3 plants-11-03471-t003:** Correlations between CTDs at three growth stages with the yield traits of the three growing seasons.

Growing Season	Traits	GPS	SN	TKW	GY	BM	HI
2016–2017 (Normal)	HD	0.04	−0.112	0.234 **	0.102	−0.065	0.232 **
	EF	−0.042	−0.193 *	0.066	−0.073	−0.238 **	0.184 *
	MF	−0.1	−0.1	0.059	0.034	−0.074	0.147 *
2017–2018 (Freezing)	HD	−0.167	0.054	−0.136	−0.039	0.102	−0.218 **
	EF	−0.023	−0.024	0.094	−0.021	0.096	−0.155 *
	MF	−0.059	0.076	0.021	−0.002	0.063	−0.071
2018–2019 (Drought)	HD	−0.279 **	−0.05	0.012	0.129	0.031	0.198 **
	EF	−0.102	0.086	−0.114	0.116	0.150 *	−0.122
	MF	−0.284 **	−0.165 *	0.072	0.08	−0.055	0.278 **

Note: The correlations were estimated using the Pearson correlations from the SPSS; * and ** indicated the correlations significant for *p* < 0.05 and *p* < 0.01, respectively. GPS: grains per spike; SN: spike number per m^2^; TKW: 1000-kernel weight; GY: grain yield; BM: biomass; HI: harvest index. HD: heading; EF: early filling; MF: middle filling.

**Table 4 plants-11-03471-t004:** Multiple comparisons of CTD and structural traits among the groups of wheat varieties based on clustering the CTDs at the three growth stages and canopy structural traits.

Growing Season	Cluster	No. of Varieties	CTD (°C)			SL (cm)	DSL (cm)	PL (cm)	PH (cm)	LFL (cm)	WFL (cm)
			HD	EF	MF						
2016–2017 (Normal)	1	85	−1.6 ± 0.81 a	−0.89 ± 0.64 a	−1.9 ± 1.24 a	9.7 ± 0.84 a	10 ± 2.1 b	27.1 ± 2.6 b	84.5 ± 5.2 b	19.3 ± 2.2 b	2.0 ± 0.15 a
	2	75	−4.0 ± 1.1 b	−3.0 ± 1.3 b	−4.6 ± 1.0 b	9.5 ± 0.83 a	9.4 ± 2.1 b	26.4 ± 2.7 b	82.7 ± 6.8 b	19.8 ± 2.3 b	1.9 ± 0.2 a
	3	26	−4.2 ± 1.2 b	−2.8 ± 1.2 b	−5 ± 0.89 b	9.0 ± 1 b	14.4 ± 4 a	32.7 ± 4.3 a	97.7 ± 11.9 a	21.5 ± 3.3 a	1.7 ± 0.2 b
2017–2018 (Freezing)	1	18	−2.4 ± 0.77 a	−2 ± 0.46 a	−1.2 ± 1.2 a	8.8 ± 1.1	14.3 ± 3.7 a	31.1 ± 3.9 a	79.1 ± 8.3 a	16.0 ± 2.1 a	1.5 ± 0.19 c
	2	86	−3.2 ± 0.71 b	−2 ± 0.54 a	−1.9 ± 0.88 b	8.7 ± 0.63	9.8 ± 2 b	24.3 ± 2.3 b	67.3 ± 4.6 b	13.9 ± 1.5 b	1.7 ± 0.16 b
	3	82	−3.9 ± 0.58 b	−2.8 ± 0.51 b	−2.3 ± 0.59 c	9.1 ± 1.4	7.3 ± 2.6 c	21.8 ± 2.6 c	63.0 ± 6.1 c	14.3 ± 1.7 b	1.8 ± 0.16 a
2018–2019 (Drought)	1	84	−0.64 ± 0.76 a	−5.4 ± 0.57 a	−1.1 ± 0.76 a	8.7 ± 0.82 a	6.0 ± 2 b	22.0 ± 2.1 b	69.6 ± 5.3 a	17.4 ± 2.2 a	1.9 ± 0.16 a
	2	14	−1.7 ± 0.76 b	−5.2 ± 0.89 a	−3.0 ± 0.72 b	8.9 ± 0.91 a	14.5 ± 3.3 a	32.8 ± 2.8 a	93.2 ± 10.7 c	18.4 ± 1.6 a	1.7 ± 0.18 b
	3	88	−2.3 ± 0.9 c	−6.0 ± 0.77 b	−2.5 ± 0.75 b	8.0 ± 0.76 b	7.1 ± 2.3 b	22.9 ± 3.4 b	75.4 ± 7.8 b	15.2 ± 2.3 b	1.7 ± 0.18 b

Note: The data are shown as the mean ± standard deviation. The clustering was conducted with the CTDs at the three growth stages and structural traits for each growing season using the Ward method. Then, multiple comparisons were conducted among different clusters using the Duncan method. Different lowercase letters indicate differences significant at the *p* < 0.05 level. WFL: width of flag leaf; LFL: length of flag leaf; SL: spike length; PH: plant height; PL: peduncle length; DSL: distance from spike to leaves; HD: heading; EF: early filling; MF: middle filling.

**Table 5 plants-11-03471-t005:** Multiple comparisons of yield traits among the clusters of varieties based on CTDs at the three stages and structural traits for each growing season.

Growing Season	Cluster	GY (kg/hm^−2^)	BM (kg/hm^−2^)	SN (m^−2^)	TKW (g)	HI
2016–2017 (Normal)	1	9406.9 ± 1895.4 a	22348.8 ± 4156.3	702 ± 121.6 b	39.5 ± 4.7 a	0.42 ± 0.04 a
	2	8777.7 ± 1538 ab	21158 ± 3303.5	665.1 ± 110.8 b	38.7 ± 4.6 a	0.41 ± 0.05 a
	3	8047.5 ± 2462.4 b	22241.2 ± 4817	762.9 ± 97 a	35.7 ± 5.5 b	0.35 ± 0.06 b
2017–2018 (Freezing)	1	6893.7 ± 1149	15316.9 ± 2241.3	382.5 ± 121.7 a	44.9 ± 5.9 b	0.45 ± 0.04
	2	6946 ± 1066.9	15316.9 ± 2315.8	347.5 ± 64.8 ab	48.2 ± 4.1 a	0.46 ± 0.05
	3	6983.3 ± 887.8	14840.3 ± 2099.8	338 ± 73.6 b	48.5 ± 4.6 a	0.47 ± 0.04
2018–2019 (Drought)	1	7297.9 ± 1259.6	14176.1 ± 2551.4 b	394.1 ± 78.9 b	48.4 ± 3.8	0.51 ± 0.05 a
	2	7789.8 ± 1783.9	16573.3 ± 4045 a	495.2 ± 97.4 a	46.6 ± 6.2	0.46 ± 0.03 b
	3	7076.3 ± 1405.5	14319.6 ± 2880.9 b	424.7 ± 82.6 b	47.4 ± 4.8	0.49 ± 0.04 a

Note: The data are shown as the mean ± standard deviation. The clustering was conducted with the CTDs at the three growth stages and structural traits for each growing season using the Ward method. Then, multiple comparisons were conducted among different clusters using the Duncan method. Different lowercase letters indicate the differences significant at the *p* < 0.05 level. SN: Spike number per m^2^; TKW: 1000-kernel weight; GY: Grain yield; BM: Biomass; HI: Harvest index.

## Data Availability

Not applicable.

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
