# Peer review of "Field Evaluation of Wheat Varieties Using Canopy Temperature Depression in Three Different Climatic Growing Seasons"

_plants, 2022, doi:10.3390/plants11243471_

Round 1

Reviewer 1 Report

The research work is interesting and falls within the scope of the journal. The authors have done research with the title “Field Evaluating of Wheat Varieties using Canopy Temperature Depression in Different Climatic Growing Seasons”. There is well consistency between the title of the article and the results and their interpretation. However, improvements like sentence structure and short sentences would make the manuscript more effective. The article dataset certainly contains constructive information for the scientific community.

The following points may be addressed by the Authors to enhance the worth of the paper. After the addition of these suggestions, I will recommend this manuscript for further processing. I will never recommend publishing this manuscript in the current write-up.

Abstract

The abstract part looks good but it is too long. Please concise it and describe the introduction, treatments and some important results. I will suggest describing the main treatment and its results at the end of the abstract.

Introduction

The write-up of the introduction part is good. The authors need to provide data about a reduction in the yield of bread wheat due to temperatures. I will suggest giving data about wheat grain composition, temperature effects on grains and annual production in the world. The authors also need to improve the sentences mentioned in specific comments.

The materials and methods are well presented. However, it needs thorough improvement in sentence structure in proofreading.

Results and discussion

The description of results and discussion part are also well. However, the author needs to justify his study results in a proper way by giving proper reasons and recent studies references. The author has used too much old references and citations.

References

There are many references cited that are too old studies. The authors need to replace them with the latest studies.

Specific Comments

Line 96-99- The sentence is long. Divide it into two sentences and improve sentence structure.

Line 100-105- The sentence is not understandable. Concise it and improve sentence structure.

 Line 120-121- On which basis 750 kg/ha fertilizer was applied and on which stages were applied?

Line-205-207- Replace the reference with the latest study.

Line 220-222- The references cited by the author are too old. I will suggest replacing it with some latest study references.

Line 240-241- The authors need to cite some latest study references.

In the reference part, the author has cited too much old references. I will suggest replacing them with some latest study references.

Author Response

Dear reviewer,  

Thank you very much for giving us an opportunity to revise our manuscript. We appreciate you very much for the positive and constructive comments, which are valuable and helpful for improving the quality of our manuscript. We have made revisions that are tracked and marked . We have tried best to make revisions to ensure better manuscript quality, and we also revised the manuscript thoroughly to make it clearer and more precise. Please find the revised version attached, which we would like to submit for your consideration.

The research work is interesting and falls within the scope of the journal. The authors have done research with the title “Field Evaluating of Wheat Varieties using Canopy Temperature Depression in Different Climatic Growing Seasons”. There is well consistency between the title of the article and the results and their interpretation. However, improvements like sentence structure and short sentences would make the manuscript more effective. The article dataset certainly contains constructive information for the scientific community.

The following points may be addressed by the Authors to enhance the worth of the paper. After the addition of these suggestions, I will recommend this manuscript for further processing. I will never recommend publishing this manuscript in the current write-up.

Abstract

The abstract part looks good but it is too long. Please concise it and describe the introduction, treatments and some important results. I will suggest describing the main treatment and its results at the end of the abstract.

Response: We have revised the abstract to make it more concise and to the point according to your requirements.

Introduction

The write-up of the introduction part is good. The authors need to provide data about a reduction in the yield of bread wheat due to temperatures. I will suggest giving data about wheat grain composition, temperature effects on grains and annual production in the world. The authors also need to improve the sentences mentioned in specific comments.

Response: Agree, we have added the data of this part to the introduction. Please see lines 107-110 in the revised manuscript.

The materials and methods are well presented. However, it needs thorough improvement in sentence structure in proofreading.

Response: Agree, it was revised accordingly.

Results and discussion

The description of results and discussion part are also well. However, the author needs to justify his study results in a proper way by giving proper reasons and recent studies references. The author has used too much old references and citations.

Response: Agree, revised accordingly, and many new references have been replaced to justify the research results.

References

There are many references cited that are too old studies. The authors need to replace them with the latest studies.

Response: Agree, it was revised accordingly.

Specific Comments

Line 96-99- The sentence is long. Divide it into two sentences and improve sentence structure.

Response: Agree, it was revised accordingly. Please see lines 211-214 in the revised manuscript.

Line 100-105- The sentence is not understandable. Concise it and improve sentence structure.

Response: Agree, it was revised accordingly. Please see lines 208-211 in the revised manuscript.

Line 120-121- On which basis 750 kg/ha fertilizer was applied and on which stages were applied?

Response: Sorry for not clear stated, the fertilization was the same for each year, and it based on local practice in wheat production. The fertilizers are applied before sowing during preparing the land. 

We have added in the revised manuscript, please see lines 275-276 in the revised manuscript.

Line-205-207- Replace the reference with the latest study.

Response: Agree, it was revised accordingly.

Line 220-222- The references cited by the author are too old. I will suggest replacing it with some latest study references.

Response: Agree, it was revised accordingly.

Line 240-241- The authors need to cite some latest study references.

Response: Agree, it was revised accordingly.

In the reference part, the author has cited too much old references. I will suggest replacing them with some latest study references.

Response: Agree, we have added new references to replace the old ones accordingly.

Reviewer 2 Report

Field Evaluating of Wheat Varieties using Canopy Temperature Depression in Different Climatic Growing Seasons is an interesting study. However following improvements are needed:

1. Abstract is ot upto mark t should be written with rationale, method, results and proper recommendations.

2. In introduction avoid repetition and add hypothesis and suggestions

3. Results should be written by mentioning high ad low.

4. Relate discussions with logics by using own data.

5. Add clear recommendations for all i.e. researchers, policy makers and farmers etc.

Author Response

Dear reviewer,  

Thank you very much for giving us an opportunity to revise our manuscript. We appreciate you very much for the positive and constructive comments, which are valuable and helpful for improving the quality of our manuscript. We have made revisions that are tracked and marked . We have tried best to make revisions to ensure better manuscript quality, and we also revised the manuscript thoroughly to make it clearer and more precise. Please find the revised version attached, which we would like to submit for your consideration.

Field Evaluating of Wheat Varieties using Canopy Temperature Depression in Different Climatic Growing Seasons is an interesting study. However following improvements are needed:

  1. Abstract is ot upto mark t should be written with rationale, method, results and proper recommendations.

Response: Agree, we have modified the abstract according to your requirements. Please see lines 11-66 in the revised manuscript.

  1. In introduction avoid repetition and add hypothesis and suggestions

Response: Agree, we have revised the introduction and added hypothesis and suggestions.

Please see lines 160-211 in the revised manuscript.

  1. Results should be written by mentioning high and low.

Response: Agree, we have modified all relevant words in the results as suggested.

  1. Relate discussions with logics by using own data.

Response: Agree, we have modified the discussion accordingly as suggested.

Please see line 526-567, 569-582, 618-631,633-763 in the revised manuscript.

  1. Add clear recommendations for all i.e. researchers, policy makers and farmers etc.

Response: Agree, it was revised accordingly, please see line 915-920 in the revised manuscript.

Based on this study, it was found that CTD could be an index for evaluating wheat germplasm in different environments, but its characteristics varied under different climatic conditions. Under normal conditions, the materials with high CTD had higher photosynthetic capacity at the early filling stage and yield traits; Under drought condition, the materials with high CTD had better photosynthetic traits, but materials with moderate CTD had higher yield traits; while in spring freezing condition, though CTD was hard for yield traits, it still could reflect the canopy structural traits and photosynthetic capacity to some extent. In a word, these recommendations might be useful for using CTD as a selection index under specific climatic conditions.

Reviewer 3 Report

The current study entitled “Field Evaluating of Wheat Varieties using Canopy Temperature Depression in Different Climatic Growing Seasons” is good. For a better understanding in-depth, it is a need for time to work on this topic. Furthermore, achieving potential benefits by using current technology depends on extensive research work for more exploration. Although the experiment is well organized, I suggest a major revision due to the following deficiencies.

Major Concerns

Title

  • The title of this study doesn't reveal anything novel. The title must succinctly convey the value addition that the authors produced in their previous work in order to catch the reader's attention. Please update this if the authors uncover any previously unreported additional findings in the current study.

Abstract

  • There is no systematic abstract. i.e., Please incorporate a subject introduction, issue description, justification for choosing the technology employed in the study at hand, knowledge gap to be filled, methodology in a few sentences, standout findings, and a conclusion.
  • In the abstract, please state the necessity for more research.
  • Please identify and declare a single, well-defined issue source that the present study is attempting to solve..
  • Give a logical explanation for why the current tactic—Field Evaluating of Wheat Varieties Using Canopy Temperature Depression—was chosen. What possible advantages may result from the current research?
  • No quantitative information is offered in result part. When explaining the findings, please provide some numbers.
  • I was unable to draw any firm conclusions from the existing research. Please provide more details on the authors' conclusions.
  • In the statement that follows, please highlight the knowledge gap that was filled, the prospective beneficiaries, and the suggestions. This statement All these results suggested that CTD would be combined with canopy structural traits to be used as a selection index for wheat grown in different climatic conditions but is more suitable for normal and drought conditions is very general. It is not a conclusion.
  • In one line, provide perspective for the future. Declare at least one best outcome.
  • As per standard suggestions, please avoid using title words as keywords.

Introduction

  • Include a unique statement at the conclusion. What new findings or correlations have authors made in this study vs earlier ones?
  • Where is the hypothesis that the present study tests? In the introduction to the current study, I couldn't discover any hypotheses.
  • Would you please give a single line about the knowledge gap your research has covered along with the SMART (specific, measurable, achievable, realistic, and time-specific) hypothesis statement?
  • No novelty statement of the study is provided at the end of the study. Please provide that.
  • Without hypothesis, it is very difficult to check either aim of the study were achieved or not. Please provide that so as I can check results and discussion part.

Author Response

Dear reviewer,  

Thank you very much for giving us an opportunity to revise our manuscript. We appreciate you very much for the positive and constructive comments, which are valuable and helpful for improving the quality of our manuscript. We have made revisions that are tracked and marked . We have tried best to make revisions to ensure better manuscript quality, and we also revised the manuscript thoroughly to make it clearer and more precise. Please find the revised version attached, which we would like to submit for your consideration.

The current study entitled “Field Evaluating of Wheat Varieties using Canopy Temperature Depression in Different Climatic Growing Seasons” is good. For a better understanding in-depth, it is a need for time to work on this topic. Furthermore, achieving potential benefits by using current technology depends on extensive research work for more exploration. Although the experiment is well organized, I suggest a major revision due to the following deficiencies.

Major Concerns

Title

The title of this study doesn't reveal anything novel. The title must succinctly convey the value addition that the authors produced in their previous work in order to catch the reader's attention. Please update this if the authors uncover any previously unreported additional findings in the current study.

Response: Agree, we have added “three” to reflect that this work involved three growing seasons with varied climatic conditions (normal, spring-freezing and drought), as most of the previous works were conducted in one or two environments, such as drought or heat.

Abstract

There is no systematic abstract. i.e., Please incorporate a subject introduction, issue description, justification for choosing the technology employed in the study at hand, knowledge gap to be filled, methodology in a few sentences, standout findings, and a conclusion.

Response: Agree, we have modified the abstract, please see line 11-66 in the revised manuscript.

In the abstract, please state the necessity for more research.

Response: Agree, we have added this part to the abstract, please see line 11-15 in the revised manuscript.

During breeding progress, the selection efficiency needs to be improved by applying the proper selection index, as there are lots of traits to be investigated for screening the elite or excellent wheat varieties or lines from a large number of materials, which cost a lot of labor and time, and the efficiency is low. Moreover, different climatic conditions will bring more complex and unpredictable situations, and make the screening more difficult. Therefore, this study was to evaluate the capability of CTD as an index for evaluating wheat germplasm under field conditions under three different climatic conditions, as CTD is easy to measure for a large number of materials, and could reflect the canopy and photosynthetic traits, which are complex and difficult to be measured. Finally, to propose the strategy for the proper and efficient application of CTD as an index in breeding programs.

Please identify and declare a single, well-defined issue source that the present study is attempting to solve. 

Response: Agree, the present study is attempting to clarify the capability of CTD as an index in evaluating main canopy structural traits, photosynthetic traits and yield traits of wheat under the changing climatic conditions, and to improving selection efficiency in wheat breeding program. Therefore, the issue that this study tries to solve is: how to screen and identify excellent wheat germplasm under different climatic conditions using CTD as a substitute index.

We have added this part in the revised manuscript, please see line 208-211.

Give a logical explanation for why the current tactic—Field Evaluating of Wheat Varieties Using Canopy Temperature Depression—was chosen. What possible advantages may result from the current research?

Response: In many previous studies, CTD has been suggested to be a reliable index for evaluating wheat germplasm for the important, complex traits such as photosynthesis and yield, under drought and heat conditions. In addition, the CTD is easy to measure with a relative cheaper equipment, such as the infrared thermometer, in a short time, which can meet the requirements of efficient and rapid measurement for large quantities of materials. In the current study, as three climatic conditions occurred in the three growing seasons, which provide an opportunity to investigate the usability of CTD in different conditions.

No quantitative information is offered in result part. When explaining the findings, please provide some numbers. 

Response: Agree, it was revised accordingly.

I was unable to draw any firm conclusions from the existing research. Please provide more details on the authors' conclusions. 

Response: Agree, we have added more details on the conclusions, please see line 915-920 in the revised manuscript.

In the statement that follows, please highlight the knowledge gap that was filled, the prospective beneficiaries, and the suggestions. This statement All these results suggested that CTD would be combined with canopy structural traits to be used as a selection index for wheat grown in different climatic conditions but is more suitable for normal and drought conditions is very general. It is not a conclusion.

Response: Agree, it was revised accordingly, please see line 915-920 in the revised manuscript.

In one line, provide perspective for the future. Declare at least one best outcome.

Response: Agree, it was revised accordingly, please see line 915-920 in the revised manuscript.

CTD could be an index for evaluating wheat germplasm for canopy structural traits, photosynthetic capability under different climatic conditions, and large quantities of wheat germplasm could be investigated quickly.

As per standard suggestions, please avoid using title words as keywords.

Response: Agree, it was revised accordingly.

Introduction

Include a unique statement at the conclusion. What new findings or correlations have authors made in this study vs earlier ones?

Response: Agree, it was revised accordingly, please see line 915-916 in the revised manuscript. Based on this study, it was found that CTD could be an index for evaluating wheat germplasm in different environments, but its characteristics varied under different climatic conditions.

Where is the hypothesis that the present study tests? In the introduction to the current study, I couldn't discover any hypotheses.

Response: Agree, the hypothesis is that under different climatic conditions, CTD at the key growth stage can be used as an index for evaluating the photosynthetic and yield characters of a large number of wheat germplasm, but what it reflects might be different under different conditions.

We have added this in the introduction, please see line 160-208 in the revised manuscript.

Would you please give a single line about the knowledge gap your research has covered along with the SMART (specific, measurable, achievable, realistic, and time-specific) hypothesis statement?

Response: Agree, it was revised accordingly, please see line 216-217 in the revised manuscript.

No novelty statement of the study is provided at the end of the study. Please provide that.

Response: Agree, the innovation of this study is that the CTD could be used as an index to screen good wheat varieties under specific climatic conditions, for the photosynthetic parameters and some yield traits.

Please see line 933-934 in the revised manuscript.

Without hypothesis, it is very difficult to check either aim of the study were achieved or not. Please provide that so as I can check results and discussion part.

Response: Agree, it was revised accordingly, please see line 160-208 in the revised manuscript.

Round 2

Reviewer 3 Report

Dear Authors 

I am satisfied with all the changes made in the manuscript. However, I still request you kindly check some language mistakes in the manuscript.

Regards

Author Response

Dear Reviewer:

Thank you very much for giving us an opportunity to check our manuscript.

We have completely checked and revised the language of this manuscript according to the comment by you, which are marked up using the “Track Change” function. 

Yours sincerely

Yin-Gang Hu, PhD
